# Transcriptomic Characterization of Genes Regulating the Stemness in Porcine Atrial Cardiomyocytes during Primary In Vitro Culture

**DOI:** 10.3390/genes14061223

**Published:** 2023-06-04

**Authors:** Rut Bryl, Mariusz J. Nawrocki, Karol Jopek, Mariusz Kaczmarek, Dorota Bukowska, Paweł Antosik, Paul Mozdziak, Maciej Zabel, Piotr Dzięgiel, Bartosz Kempisty

**Affiliations:** 1Section of Regenerative Medicine and Cancer Research, Natural Sciences Club, Faculty of Biology, Adam Mickiewicz University, Poznań, 61-614 Poznan, Poland; rutbryl@gmail.com; 2Department of Anatomy, Poznan University of Medical Sciences, 60-781 Poznan, Poland; mjnawrocki@ump.edu.pl; 3Department of Histology and Embryology, Poznan University of Medical Sciences, 60-781 Poznan, Poland; karoljopek@ump.edu.pl; 4Department of Cancer Immunology, Chair of Medical Biotechnology, Poznan University of Medical Sciences, 61-866 Poznan, Poland; markacz@ump.edu.pl; 5Gene Therapy Laboratory, Department of Cancer Diagnostics and Immunology, Greater Poland Cancer Centre, 61-866 Poznan, Poland; 6Department of Diagnostics and Clinical Sciences, Institute of Veterinary Medicine, Nicolaus Copernicus University in Torun, 87-100 Torun, Poland; dbukowska@umk.pl; 7Department of Veterinary Surgery, Institute of Veterinary Medicine, Nicolaus Copernicus University in Torun, 87-100 Torun, Poland; pantosik@umk.pl; 8Prestage Department of Poultry Science, North Carolina State University, Raleigh, NC 27695, USA; pemozdzi@ncsu.edu; 9Physiology Graduate Faculty, North Carolina State University, Raleigh, NC 27695, USA; 10Department of Human Morphology and Embryology, Division of Histology and Embryology, Wroclaw Medical University, 50-368 Wroclaw, Poland; maciej.zabel@umw.edu.pl (M.Z.); piotr.dziegiel@umw.edu.pl (P.D.); 11Division of Anatomy and Histology, University of Zielona Góra, 65-046 Zielona Góra, Poland; 12Department of Human Morphology and Embryology, Division of Anatomy, Wroclaw Medical University, 50-367 Wroclaw, Poland; 13Department of Obstetrics and Gynaecology, University Hospital and Masaryk University, 62500 Brno, Czech Republic

**Keywords:** cardiomyocytes, porcine cardiac muscle, long-term in vitro cell culture, stemness markers, transcriptomic analysis

## Abstract

Heart failure remains a major cause of death worldwide. There is a need to establish new management options as current treatment is frequently suboptimal. Clinical approaches based on autologous stem cell transplant is potentially a good alternative. The heart was long considered an organ unable to regenerate and renew. However, several reports imply that it may possess modest intrinsic regenerative potential. To allow for detailed characterization of cell cultures, whole transcriptome profiling was performed after 0, 7, 15, and 30 days of in vitro cell cultures (IVC) from the right atrial appendage and right atrial wall utilizing microarray technology. In total, 4239 differentially expressed genes (DEGs) with ratio > abs |2| and adjusted *p*-value ≤ 0.05 for the right atrial wall and 4662 DEGs for the right atrial appendage were identified. It was shown that a subset of DEGs, which have demonstrated some regulation of expression levels with the duration of the cell culture, were enriched in the following GO BP (Gene Ontology Biological Process) terms: “stem cell population maintenance” and “stem cell proliferation”. The results were validated by RT-qPCR. The establishment and detailed characterization of in vitro culture of myocardial cells may be important for future applications of these cells in heart regeneration processes.

## 1. Introduction

The past three decades have brought progress in cardiovascular medicine leading to a significant decrease in mortality due to acute cardiovascular syndromes in developed countries [1]. Still, in many cases, myocardial injury leads to the development of heart failure, “a complex clinical syndrome that results from any structural or functional impairment of ventricular filling or ejection of blood” [2]. Heart failure (HF) has an annual prevalence of approximately 37.7 million people [3]. Survival of HF patients varies from 3 to 5 years, on average, making their prognosis poorer than for most cancers [4].

There is a need to develop more optimal methods for the treatment of heart failure. The heart was considered an organ unable to regenerate and renew. However, there is now evidence suggesting that the heart may have some ability for self-repair, especially in response to exploitation/physiological stress [5,6,7,8,9,10]. Therapies based on cells driving heart regeneration may become a promising new strategy for current disease management.

Cardiac progenitor cells (CPCs) are potentially a heterogenous group of cells which activate in response to an injury and contribute to cardiomyocytes’ replenishment. They are localized in different heart regions (atria, ventricles, epicardium, or pericardium) [11]. Many studies in the past two decades have focused on the identification and isolation of this cell population in adult mammalian hearts. Their characteristic properties include clonogenicity, self-renewal, differentiation into several cell types of cardiac lineage, expression of specific transcription factors—Isl-1, Nkx2.5, MEF2C, and GATA-4—and stemness markers such as Oct3/4, Bmi-1, and Nanog. Intramyocardial transplantation of such cells leads to a reduction of scar size and the preservation of left ventricular function in preclinical myocardial infarction models [11,12,13]. These subpopulations include c-kit^pos^ CPCs [14,15,16], cardiosphere-derived cells (CDCs) [17,18,19,20], cardiac side population cells (SP) [21,22,23], Sca-1^pos^ CPCs (stem cell antigen-1) [24,25], epicardium-derived progenitor cells (EPDCs) [26,27], cardiac colony-forming unit fibroblasts (cCFU-Fs) [28,29], and Islet-1^pos^ cells [30,31,32].

Two types of adult cardiac progenitor cells have reached clinical testing [33,34]. It is noteworthy that results of CADUCEUS, a clinical study of cardiosphere-derived cells, showed a reduction of infarct size, increased viable heart mass, and regional contractility in the group administered with autologous CDCs compared to the group that received standard treatment. Nevertheless, there was no significant difference in left ventricular (LV) ejection fraction between the groups [34].

A significant challenge to employ cardiac stem cells is the small pool of these cells. Furthermore, the precise role of the majority of CPCs in the adult heart is unknown [11]. As a consequence, molecular characterization of these cell populations and the elucidation of genetic and epigenetic mechanisms which dictate the cells’ phenotype and fate is crucial for further progress.

The aim of this work was to establish a methodology for the identification of presumed cardiac stem cells, as well as for primary in vitro cultures’ conditions of cardiomyocytes isolated from two different fragments of porcine (*Sus scrofa f. domestica*) heart, namely, the right atrial appendage and right atrial wall. It is suggested that there may reside a specific cell population with regenerative potential within porcine cardiac muscle. The establishment and detailed characterization of in vitro culture of cardiac muscle cells may be important for future applications of these cells in heart regeneration processes.

## 2. Materials and Methods

### 2.1. Animals

For this study, a total of 25 pubertal crossbred Landrace gilts bred on a commercial local farm were used. The gilts had a mean age of 155 days (range 140–170 days), and the mean weight was 100 kg (95–120 kg). All of the animals were housed under identical conditions and fed the same forage (depending on age and reproductive status).

This research related to animal use has complied with all the relevant national regulations and institutional policies for the care and use of animals. As the research material is usually disposed of after slaughter, being a remnant by-product, no Ethical Committee approval was needed for this study.

### 2.2. Cell Isolation

After slaughter, porcine hearts were transported to the laboratory on ice within 40 min of harvest. Next, two heart fragments were excised from the right atrial appendage and right atrial wall. The excised tissue was washed in 10% povidone–iodine solution (Betadine; EGIS, Warsaw, Poland) at room temperature, twice in 0.9% sodium chloride solution (NATRIUM CHLORATUM 0,9% FRESENIUS, Fresenius Kabi, Warsaw, Poland) at room temperature, and eventually in cold (kept at 4 °C) Dulbecco’s Phosphate Buffered Saline (D-PBS; SIGMA-ALDRICH, St Louis, MO, USA, Merck KGaA, Darmstadt, Germany) (137 mM NaCl, 27 mM KCl, 10 mM Na_2_HPO_4_, 2 mM KH_2_PO_4_, pH 7.4) supplemented with 2% antibiotics/antimycotics (100 U/mL penicillin G sodium, 100 μg/mL streptomycin sulfate, and 0.25 μg/mL amphotericin B) (Antibiotic Antimycotic Solution (100×); SIGMA-ALDRICH, St. Louis, MO, USA, Merck KGaA, Darmstadt, Germany). Next, tissue was subjected to two-step mincing in Petri dishes using sterile tools. Firstly, the heart fragments were placed in a droplet of cold Dulbecco’s Modified Eagle Medium (DMEM; SIGMA-ALDRICH, St. Louis, MO, USA, Merck KGaA, Darmstadt, Germany), cut into cubes with sides of ~5 mm, and transferred to 50 mL conical tubes for intensive washing with cold D-PBS. Washed fragments were then minced into ~1 mm^3^ fragments, also in a droplet of cold DMEM. Each tissue fragment was incubated in 10 mL of collagenase type II solution (collagenase type II; SIGMA-ALDRICH, St. Louis, MO, USA, Merck KGaA, Darmstadt, Germany) in DMEM (c = 1 mg/mL) with shaking (100 rpm) for 40 min at 37 °C (Orbital Shaker with a Heating Module and Incubator Hood, Incubator 1000/Polymax 1040 model; Heidolph Instruments GmbH & Co. KG, Schwabach, Germany). After digestion, the solution was filtered through an autoclavable stainless steel cell sieve with steel mesh and subsequently through 70 μm nylon cell strainers. The filtrate was transferred to 15 mL conical tubes and centrifuged (5 min, 200× *g*, room temperature) (Centrifuge 5810R, Eppendorf AG, Hamburg, Germany). After centrifugation, the supernatant was discarded and the pellet was resuspended in 5 mL of cold D-PBS. The centrifugation was repeated and the pellet resuspended in 1 mL of DMEM/F12 supplemented (cardiomyocytes culture medium) at 37 °C consisting of DMEM/Nutrient Mixture F-12 Ham (DMEM/F12; SIGMA-ALDRICH, St. Louis, MO, USA, Merck KGaA, Darmstadt, Germany), 20% Fetal Bovine Serum (FBS; SIGMA-ALDRICH, St. Louis, MO, USA, Merck KGaA, Darmstadt, Germany), 10% Horse Serum (HS; SIGMA-ALDRICH, St. Louis, MO, USA, Merck KGaA, Darmstadt, Germany) and 1% antibiotics/antimycotics (100 U/mL penicillin G sodium, 100 μg/mL streptomycin sulfate, and 0.25 μg/mL amphotericin B) (Antibiotic Antimycotic Solution (100×); SIGMA-ALDRICH, St. Louis, MO, USA, Merck KGaA, Darmstadt, Germany). Cells were then seeded in 25 mL bottles (at a final volume of 4 mL of medium).

### 2.3. Long-Term Primary Cell Culture

Cells were cultured at 37 °C in a humidified atmosphere of 5% CO_2_ in 25 mL bottles for a maximum of 30 days.

Once the cultures reached 70–80% confluency, they were passaged by washing with 3 mL of D-PBS, digested with 3 mL of 0.025% trypsin/EDTA (Trypsin-EDTA solution (0.25%), SIGMA-ALDRICH, St. Louis, MO, USA, Merck KGaA, Darmstadt, Germany) at 37 °C for 3–5 min, neutralized by the addition of 6 mL of DMEM supplemented with 10% FBS, 4 mM L-glutamine solution (L-glutamine solution (200 mM), SIGMA-ALDRICH, St. Louis, MO, USA, Merck KGaA, Darmstadt, Germany) and 1% antibiotics/antimycotics (100 U/mL penicillin G sodium, 100 μg/mL streptomycin sulfate and 0.25 μg/mL amphotericin B) (Antibiotic Antimycotic Solution (100×); SIGMA-ALDRICH, St. Louis, MO, USA, Merck KGaA, Darmstadt, Germany), centrifuged (5 min, 200× *g*, room temperature), and resuspended in DMEM/F12 supplemented (cardiomyocytes culture medium).

The medium was changed every 3 days and the culture was observed under an inverted microscope employing relief contrast (IX73, Olympus, Tokyo, Japan) every day. Before the harvesting of cells for further experiments, photos of the culture at three time points, 7 days, 15 days, and 30 days, were also taken to observe possible morphological changes.

### 2.4. Detection of Chosen Markers Using Flow Cytometry (FC)

After reaching a proper degree of confluence, the cells were digested from the culture plate, using 0.025% trypsin/EDTA solution, and subjected to FC analysis through staining. To identify characteristic markers, the following antibodies were used: Anti-CD44 FITC (Abcam, Cambridge, UK), Anti-CD90 APC, and Anti-CD105 PE (R&D systems). For the staining of cytoplasmic proteins, firstly, cells were permeabilized and fixed in BD Perm/Wash™ buffer on ice. Next, 300 μL of BD Perm/Wash™ buffer was added to 100 μL of cells in solution. The samples were subsequently centrifuged (250× *g*, 5 min), supernatant was discarded, and cells were incubated with antibodies for 30 min on ice. Following incubation, a wash with BD Perm/Wash™ buffer was performed. Such prepared cells were analyzed with a BD FACSAria™ cytometer and FACSDiva™ software (Becton Dickinson, Franklin Lanes, NJ, USA). Cell surface markers (CD44, CD105, and CD90) were identified, then 5 μL of antibodies were added to 100 μL of cell suspension. The mix was washed twice in PBS and centrifuged (250× *g*, 5 min). The pellet, remaining after supernatant removal, was resuspended in 100 μL of PBS and subjected to acquisition in a flow cytometer.

### 2.5. RNA Extraction and Isolation

Total RNA was extracted after 7, 15, and 30 days of the cell culture, respectively. The cells were treated as if they were passaged (see Long-term primary cell culture for details) but the pellet was resuspended in 500 µL of TRI Reagent Solution (TRI Reagent^®^, SIGMA-ALDRICH, St. Louis, MO, USA, Merck KGaA, Darmstadt, Germany). The aliquots were stored in −80 °C until RNA isolation.

Total RNA was isolated according to the Chomczyński and Sacchi method [35,36]. Firstly, 100 µL of chloroform (SIGMA-ALDRICH, St. Louis, MO, USA, Merck KGaA, Darmstadt, Germany) was added, the samples were mixed by inversion and shaken for 15 sec, and then incubated for 10 min at room temperature. Subsequently, the biphasic emulsion was separated by centrifugation at 12,000× *g* for 15 min at 4 °C. The upper, aqueous phase which contained RNA was transferred to new Eppendorf tubes. Next, 250 µL of isopropanol (SIGMA-ALDRICH, St. Louis, MO, USA, Merck KGaA, Darmstadt, Germany) was added, the samples were mixed by inversion, shaken for 15 s, and incubated for 15 min at room temperature. The samples were centrifuged at 12,000× *g* for 10 min at 4 °C. Then, 1 ml of 75% ethanol solution (SIGMA-ALDRICH, St. Louis, MO, USA, Merck KGaA, Darmstadt, Germany) was added to the precipitate, the samples were vortexed for 20 s, and centrifuged at 7500× *g* for 15 min at 4 °C. The supernatant was discarded and the samples were air-dried and dissolved in 20–50 µL (depending on pellet size) of DEPC-treated water.

The spectrophotometric analysis at λ = 260 nm (NanoDrop spectrophotometer; Thermo Fisher Scientific, Waltham, MA, USA) was performed in order to assess the concentration of the samples.

### 2.6. Microarray-Based Transcriptomic Profiling

Total RNA (100 ng) from each pooled sample was subjected to two rounds of sense cDNA amplification (Ambion^®^ WT Expression Kit, Life Technologies, Carlsbad, CA, USA). The obtained cDNA was used for biotin labeling and fragmentation by Affymetrix GeneChip^®^ WT Terminal Labeling and Hybridization (Affymetrix, Santa Clara, CA, USA). Biotin-labeled fragments of cDNA (5.5 μg) were hybridized to the Affymetrix^®^ Porcine Gene 1.1 ST Array Strip (48 °C/20 h). Microarrays were then washed and stained according to the technical protocol using the Affymetrix GeneAtlas Fluidics Station. The array strips were scanned employing Imaging Station of the GeneAtlas System. Preliminary analysis of the scanned chips was performed using Affymetrix GeneAtlas^TM^ Operating Software. The quality of gene expression data was confirmed according to the quality control criteria provided by the software. The obtained CEL files were imported into downstream data analysis software.

### 2.7. Reverse Transcription-Quantitative Polymerase Chain Reaction (RT-qPCR)

An amount of 1 µg of isolated RNA diluted in PCR-grade water (up to a final volume of 8 µL) from each sample was reverse transcribed by RT2 First Strand Kit (Qiagen^®^, Hilden, Germany) according to the protocol. With the exception of incubation, samples were kept on ice. First, to eliminate the genomic DNA, 2 µL of GE (5x ×gDNA Elimination Buffer) was added to 1 µg of isolated RNA. Samples were incubated at 42 °C for 5 min. Next, the reaction mix, which included 4 µL BC3 (5× RT Buffer 3), 1 µL P2 (Primer and External Control Mix), 2 µL RE3 (RT Enzyme Mix 3), and PCR-grade water up to the final volume of 10 µL, was prepared and 2 incubations were performed: at 42 °C for 15 min, and at 95 °C for 5 min. Next, samples were cooled on ice and 91 µL of H_2_O was added to each reaction.

The validation of microarray data was/will be performed on LightCycler^®^ 96 Instrument (Roche Diagnostics GmbH, Mannheim, Germany). cDNA synthesized in reverse transcription served as a template. Primers were designed in Primer3Plus software (version 0.4.0; Whitehead Institute for Biomedical Research, Massachusetts Institute of Technology, Cambridge, MA, USA) [37,38,39] using sequences of chosen transcript variants of the genes available in the Ensembl database [40] (Table 1). The components of the reaction mix were as follows: QUANTUM EvaGreen^®^ PCR Kit (5×) (Syngen Biotech, Wroclaw, Poland) which was used as the master mix, 10 µM of oligodeoxynucleotides (SIGMA-ALDRICH, St Louis, MO, USA, Merck KGaA, Darmstadt, Germany), and PCR-grade water. Next, 9 µL of the reaction mix and 1 µL of the template were added to each respective well on a 96-well plate. The plate was sealed with a sealing foil, centrifuged at 1500 rpm 400 × *g* for 1 min, and placed in the thermocycler. The thermal profile of the reaction is presented in Table 2.

The 2^−ΔΔCT^ method for relative gene expression analysis was applied [41,42]. cDNA synthesized from RNA isolated after 0 days of cell culture served as a control. The relative expression of the studied genes was normalized against the expression of 2 reference genes—*GAPDH* and *ACTB*.

### 2.8. Bioinformatic Data Analysis and Visualization

All of the presented analyses were performed and prepared using the R programming language with the Bioconductor package [43,44]. Each CEL file was merged with a description file. In order to correct the background, normalize, and summarize results, the Robust Multiarray Averaging (RMA) algorithm was used. To determine the statistical significance of the analyzed genes, moderated t-statistics from the empirical Bayes method were computed. The obtained *p*-value was corrected for multiple comparisons using Benjamini and Hochberg’s procedure. The selection of differentially expressed genes (DEGs) was based on adjusted *p*-value ≤ 0.05 and ratio > abs(1). The DEGs list was uploaded to DAVID (Database for Annotation, Visualization, and Integrated Discovery) [45,46,47]. DAVID was used to retrieve Gene Ontology Biological Process (GO BP) terms enriched in the list of identified genes [48,49]. Significantly changed GO BP terms were defined as those enriched in at least 5 genes and for which the adjusted *p*-value ≤ 0.05. Plots were generated using the GOplot package [50].

RT-qPCR analyses and graphs were performed and prepared using Microsoft Excel (Microsoft Corporation). The results are presented as mean values ± standard deviation (SD) of two technical replicates of each reaction.

## 3. Results

### 3.1. Isolation of Cells from Two Fragments of Porcine Cardiac Muscle and Long-Term In Vitro Primary Culture

Cell isolation, establishment, and maintenance of the cell culture from two fragments, namely, the right atrial appendage and right atrial wall, was successful. The cultures were observed every 3 days up to the 30th day (Figure 1). Cell morphology was similar in all cultures. Initially, cells had rather irregular shapes and gained a spindle-like appearance with the duration of the culture. Interestingly, we observed the formation of 3D clusters as captured for the culture of cells from the right atrial wall. The choice of those fragments was based on the fact that they can be easily collected during open heart surgery for coronary artery bypass grafting and, therefore, performed experiments could be later translated into humans. Two fragments—the right atrial appendage and right atrial wall—were selected for further characterization due to the ease of collection.

### 3.2. Identification of Possible Stemness Phenotype of the Cells Isolated from the Heart Tissue

In order to enable a comparison of specific regions of the heart as potential locations of concentration of cells with presumed stem cell characteristics, CD44, CD90, and CD105 were chosen and the cells were analyzed for their presence with flow cytometry in the 7th, 15th, and 30th days of primary cultures. Results from the 7th day of cell culture derived from the right atrium show some expression of cytoplasmic CD105 (MFI fold change of 1.82). A similar situation was observed for the 15th day of cell culture (MFI fold change of 1.84), while for the 30th day, expression of cytoplasmic CD105 and CD90 was detected (MFI fold change of 2.88 and 3.12, respectively) (Figure 2). As for the right atrial appendage, cytoplasmic CD105 and CD90 was expressed in cells derived from the 7th-day culture (MFI fold change 1.89 and 2.03, respectively), while cytoplasmic CD105 was also expressed in the 15th day of culture (MFI fold change 1.95) and CD90 on the surface and in the cytoplasm were detected in the cells from the 30th-day culture (MFI fold change 11.4 and 5.08, respectively) (Figure 3). It should be noted however that the number of events—i.e., the number of cells—is relatively low (frequently below 100), except for the sample derived from the right atrial appendage in the 30th day of culture.

### 3.3. Transcriptomic Characterization of the Cardiomyocytes during Long-Term In Vitro Culture

To further understand molecular changes occurring in CMs under the influence of in vitro culture conditions, whole transcriptome profiling by Affymetrix microarray was performed. Gene expression changes between 0 and 7, 15, and 30 days of cardiomyocytes culture were studied. A total of 12,257 transcripts were examined by Affymetrix^®^ Porcine Gene 1.1 ST Array Strip. Genes were considered as differentially expressed (differentially expressed genes, DEGs) based on 2 parameters: ratio > abs |2| and adjusted *p*-value ≤ 0.05.

Followingly, biologically relevant processes in which the DEGs could be implicated were explored. For this purpose, DAVID (Database for Annotation, Visualization, and Integrated Discovery) software was utilized and enabled the extraction of gene ontology biological process terms (GO BP), which contain DEGs. DAVID searching was performed for up- and downregulated gene sets separately, and only gene sets in which the adjusted *p*-value ≤ 0.05 were selected. Two gene ontology biological process terms (GO BP) were chosen in this work: “stem cell population maintenance” and “stem cell proliferation”. The chosen sets were then subjected to a hierarchical clusterization procedure and presented as heatmaps (Figure 4). The symbols, fold changes in expression, and adjusted *p*-values for the 10 most deregulated genes at each time point in culture are shown in Appendix A. The majority of genes which belong to the GO term “stem cell maintenance” in samples derived from right atrial wall tissue are upregulated in comparison to day 0, and this tendency remains until day 30. *FGF2*, *EIF4E*, *TBX3*, *LIF*, and *EPAS1* showed a gradual increase in expression levels with the duration of culture, with the highest expression in day 30 among all the genes in this GO term. There are, however, several genes in which expression in comparison to day 0 seems unchanged or was downregulated. Similar tendencies can be observed for genes in the group “stem cell proliferation”. In this case, *TBX3* and *SOX5* were the most upregulated, while downregulation was noted for *PAX6*, *WNT2B*, *FGFR2,* and *SNAI2* (Figure 4). In the samples from the right atrial appendage, approximately half of the genes enriched in the “stem cell maintenance” GO term showed a slight increase in expression in relation to day 0, while the other half experienced initial upregulation, especially in day 7, slight in day 15, and subsequent downregulation in day 30. Similar observations can be made when analyzing the genes for the “stem cell proliferation” GO term in the same samples (Figure 4).

To check if the genes belong to more than one ontological group and to characterize these groups in a more detailed manner, the results of gene annotation enrichment analysis were visualized with the circle plot (GOCircle) (Figure 5), the relationships between genes and terms were plotted with the GOChord plotting function (Figure 6), and a heatmap of genes and terms was generated with the GOHeat function (Figure 7).

The circle plot reveals that the genes in the samples from the right atrial appendage show a general tendency of expression level upregulation. There are more genes enriched for the “stem cell proliferation” GO term than for “stem cell population maintenance”. As for the samples from the right atrial appendage, although the tendency of gene expression upregulation can also be observed, some genes in the sample from the 30th day of cell culture show decreased expression levels. Furthermore, the “stem cell proliferation” GO BP term is slightly decreased (based on the z-score).

Moving to Figure 6, for samples from the right atrial wall, only 1 of the displayed genes was present in both GO terms, and all of the genes showed greater or lesser upregulation in day 7 compared to day 0. In the samples derived from the right atrial appendage, 5 of the presented genes were enriched in both GO terms, and only 2 of the displayed genes showed a tendency to decrease.

The heatmap presented in Figure 7 shows common differentially expressed genes for both GO terms in the 7th vs. 0 day of the culture. Genes *SFRP2* and *PRRX1* are the most highly expressed for both the right atrial wall and right atrial appendage.

The RT-qPCR (reverse transcription-quantitative polymerase chain reaction) technique was also utilized to validate the chosen results of high-throughput transcriptome profiling. The results are demonstrated as a bar chart (Figure 8).

## 4. Discussion

Despite recent progress in cardiology having caused a significant decrease in CVD (cardiovascular disease)-related deaths, the number of heart failure cases has increased [1]. There has been an increasing interest in regenerative medicine that would allow finding a cellular replacement for the cardiomyocytes lost during the disease [13].

In recent years, the dogma stating that the adult heart lacks any regenerative potential has been questioned by the evidence that cardiac muscle cells possess some small self-renewal capacity over the mammalian lifetime. Heart precursor cells have been described in adult hearts of several mammalian species. However, there is no unified methodology for the isolation of cardiac stem cells [14,28,51,52,53].

Primary cultures of cells from two of the porcine heart tissue fragments have been successfully established and, as evident by results from a previous study of the group, namely, the presence of α-MHC and α-actinin as well as GATA-4 in the cells analyzed by flow cytometry, CM from the right atrial wall and right atrial appendage have been isolated [14,54]. In the last two decades, there has been a significant increase in the number of types of cardiac progenitor cells reported. These cells have been identified and isolated based on the expression of a whole variety of markers on their surface which overlap to some extent and are highly mixed. Here, CD90, CD44, CD105, and GATA-4 have been proposed as putative stemness markers [54] and a small population of cardiac muscle-derived cells expressing these markers has been identified.

CD90 (THY1) has been described as a marker of a variety of stem cells, including mesenchymal stem cells [55,56,57]. As for other cells, CD90 was described as characteristic for endothelial cells, smooth muscle cells, or fibroblasts, among others [58,59,60]. According to some reports, CD90 can be present on cardiac progenitors [28,61,62,63,64], while others have shown that only CD90^neg^ CSCs can have any therapeutic role and that they generally outperform unsorted cells [65]. The flow cytometry results have revealed the presence of this marker in the cytoplasm of cells derived from the right atrial wall in the 30th day of the IVC and derived from the right atrial appendage in the 7th day of the IVC. Cells derived from the right atrial appendage in the 30th day seem to point out the presence of this marker on their cell surface. As the populations of cardiac progenitor cells are usually small, this may suggest the presence of other cell types typical for the heart. It should be noted however that the number of events in case of these samples is relatively small and thus the interpretation cannot be held fully reliable—additional experiments involving more viable cells would be required.

CD44 is a surface glycoprotein which participates in the transduction of signals important for cell migration, cell adhesion, and cell–cell interaction, and it plays a role in sensing cues from the microenvironment [66]. It interacts with several components of the extracellular matrix, including hyaluronan [67]. Similarly, to CD90, it is a canonical marker of mesenchymal stem cells [55]. There were several reports of cardiac progenitor cells which are CD44^pos^, including epicardium-derived progenitor cells and cardiac colony-forming unit fibroblasts [28,68]. According to FC, very few cells have shown CD44 expression. Conversely, it is widely known that cardiac stem cells constitute a small population of cells in the adult mammalian heart [13].

Endoglin, or CD105, is a type I transmembrane protein which serves as a BMP9/10 co-receptor and TGF-β signaling auxiliary receptor that is expressed primarily on vascular endothelial cells [69,70]. It is important for maintaining the proper structure of vasculature and is indispensable for heart development [71]. Furthermore, together with CD90 and CD44, CD105 is a marker of mesenchymal stem cells [55]. There are numerous reports of CD105 presence on cardiac progenitor cells, especially for cardiosphere-derived cells, Sca-1^pos^ CPC, and EDPCs [25,28,61,63,64]. According to our FC analysis, there is a small population of cells expressing this marker.

GATA-4 is a transcription factor which regulates early cardiac development and specification [72,73,74]. Together with ISL1, MEF2C, and NKX2-5, it is often used for the identification of cardiac phenotype in heart precursor cells [11]. GATA-4 is expressed in CPCs including Sca1^pos^ cells, Isl-1^pos^ cells, and cardiac fibroblasts [31,51,52,75]. Apart from its role in cardiac morphogenesis at the embryonic stage, GATA-4 is crucial for cardiac hypertrophy by regulating transcriptional activation of cardiac-specific genes [74,76]. BRD4-GATA4 have also been recently revealed to regulate adult cardiac metabolism [77].

FC analysis has revealed that GATA-4 was expressed in all samples except the 30D right atrial appendage (MFI fold change in the range 2–4 compared to the control) [54]. It is therefore suggested that GATA-4 could regulate the functioning of adult cardiomyocytes in culture, including their growth.

Despite several phenotypic variations of cardiac progenitor cells having been identified, the precise role and molecular identity of CPCs in the adult heart remains enigmatic. A small pool of precursor cells is also a significant challenge and underlines the importance of the research that would allow molecular profiling of CPCs. There are studies which report characterization of the mammalian cardiac progenitor cells’ transcriptome but the majority utilize different techniques than ours and aim at studying changes occurring at the embryonic stage [78,79,80]. Nevertheless, molecular changes at the transcriptomic and epigenomic level in putative adult murine heart-derived CPCs have been described [81,82]. Interestingly, in 2013, Dey et al. reported microarray-based transcriptional profiling of five cardiac (ckit^+^, Sca1^+^, and side population) and bone marrow-derived (ckit^+^ and mesenchymal stem cell) progenitors from adult mice and found out that cardiac ckit^+^ cells are the most distinct and seem to be the most primitive cell population [83]. However, the hypothesis that cells present in the bone marrow transdifferentiate to cardiomyocytes was disproven and the studies of Anversa et al., who first identified c-kit^+^ cells, are now not held reliable [84,85]. Additionally, genetic lineage tracing studies have revealed that Sca1+ cells give rise rather to endothelial cells and not myocytes [86]. The porcine cardiac transcriptome has been recently reported as updated [87] and there are several studies in which transcriptomic profiling of porcine hearts was utilized to address different questions [88,89,90]. It appears that the current study is the first analysis that aims at transcriptome profiling of porcine cardiomyocytes during long-term in vitro culture.

*SFRP2* was one of the most highly upregulated genes in cells derived from both heart fragments through the duration of the culture. For the right atrial wall, a certain decrease in expression levels with IVC time was observed; nevertheless, it remained as the 1st- or 2nd-most highly upregulated gene, while for the right atrial appendage, the expression increased in the 15th day compared to the 7th day and remained high until the end.

*SFRP2*, a secreted frizzled-related protein 2, is a member of the sFRP family which regulates both canonical and noncanonical Wnt signaling, influencing many fundamental processes such as proliferation, apoptosis, and differentiation [91]. β-catenin-dependent and β-catenin-independent Wnt signaling pathways are indispensable for heart development [92]. As for cardiomyocytes differentiation, research has shown that after the initial activation of β-catenin, required for mesoderm specification to cardiac progenitors, it is the inhibition of this molecule which allows for differentiation of these progenitors to cardiomyocytes [93]. It has also shown to be important in cardiac fibrosis as well as angiogenesis, thereby contributing to the restoration of the cardiac function after injury [94,95]. Studies on adult murine cardiac progenitor cells (Sca1^+^) have shown that Sfrp2 may promote the differentiation of such cells into CMs after injury [96]. It has also been shown that myocardial injury may induce c-kit^+^ cells to exhibit cardiomyocyte-specific gene expression and Sfrp2 enhances this effect [97], and, lastly, the same group has claimed that Sfrp2 induces cardiomyocyte differentiation in c-kit^+^ cells in vivo in health and MI, as revealed by genetic lineage tracing and functional assessment of the developed CMs [98].

Although, at the early stages of cardiomyogenesis, *SFRP2* prevents the generation of cardiac progenitors from mesoderm cells and promotes the maintenance of cells in the undifferentiated state [99], as mentioned above, there is evidence in adult cardiac progenitors that it facilitates differentiation to cardiomyocytes and, furthermore, it promotes cardiac fibrosis. As the cells in this study were isolated from adult porcine hearts, it is suspected that upregulation of this molecule had rather a negative effect on maintaining the stemness phenotype/undifferentiated state.

*PRRX1* (Paired Related Homeobox 1) was also one of the two most highly upregulated genes in samples from both heart fragments. For the right atrial wall, a situation was similar to *SFRP2*—compared to the 7th day, a certain decrease in *PRRX1* levels in the 15th and 30th days of the culture was observed, and the ratios were generally similar to *SFRP2*. For the right atrial appendage, the expression increased in the 15th day compared to the 7th day, but only slightly, and, overall, the expression levels were 2–3 times lower than for *SFRP2*.

*PRRX1* belongs to the paired family of homeobox-containing transcription factor proteins and has nuclear localization.

There are several studies characterizing the expression of *PRRX1* in the developing hearts of vertebrates, mainly in mesenchymal tissues, including in the heart and arteries [100,101,102].

*Prrx1* is involved in scar-free heart regeneration. A 2021 study by de Bakker et al. demonstrated that loss of *Prrx1b* in cryo-injured zebrafish hearts led to excessive fibrosis and a reduction of cardiomyocyte proliferation at the injury border zone. Single cell sequencing and lineage tracing have shown that *Prrx1* is expressed mostly in epicardial and epicardial-derived cells and that there is an excess of fibroblasts producing TGF-β ligand and ECM in injured prrx1b^−/−^ hearts. Mechanistically, Prrx1 promotes Ngr1 and, consequently, cardiomyocytes’ proliferation not only in in vivo zebrafish model but, noticeably, in human fetal epicardial-derived cells [103].

As a transcription factor, PRRX1 can physically interact and regulate expression of genes crucial for proper cardiac function. In a study investigating AF pathogenesis, Guo et al. showed that *SHOX2* and *ISL1*,*2* were genes that played roles in cardiac pacing and conducting systems that undergo such regulation [104].

In another 2021 study, it was revealed that deletion of a noncoding variant genomic region at 1q24 associated with downregulation of *Prrx1* leads to a gain of more myogenic phenotype in atrial cardiomyocytes. This suggests that *Prrx1* is a negative regulator of cardiac muscle lineage-specific gene expression. They also described a negative feedback regulatory loop between Prrx1 and Mef2, an important regulator of cardiomyocytes’ development and differentiation, which may indicate that this relationship is important in the maintenance of cardiomyocyte homeostasis [105].

It is difficult to unequivocally assess the potential influence of *PRRX1* on the long-term culture. Many of the previous studies have described the role of *PRRX1* in embryonic development, while here, the cells were collected from adult pigs. It seems that studies on AF seem to be the closest to the current system, as they were mostly conducted on mammalian–human or murine cells, although not porcine. *PRRX1* could potentially promote cardiomyocytes’ proliferation and control expression of genes involved in the proper functioning of cells, as well as regulate the myogenic phenotype. The culture consists of cells which express markers characteristic for cardiomyocytes based upon flow cytometry data. At the same time, the authors do not have broader transcriptomic data which would possibly reveal a profile of expression of other genes important for the determination of such a phenotype. Furthermore, it is interesting that contradictory trends in the expression of this gene have been observed with the duration of culture between samples from the right atrial appendage and right atrial wall.

The other two most deregulated genes in the presented data are *RACGAP1* and *SNAI2*. *RACGAP1*, Rac GTPase Activating Protein 1, here enriched in the “stem cell proliferation” GO BP term, is crucial for cytokinesis, namely, for the myosin contractile ring formation as well as proper attachment of the midbody to the cell membrane [106,107]. Many reports describe its role in the development and progression of different cancers, where it may serve as a proliferation and poor prognosis marker [108,109,110]. It has also been shown that RACGAP1 promotes hematopoietic stem cells’ differentiation and inhibits cell growth [111].

It has been shown that *brk1*, *nckap1*, and *wasf2*, which belong to WAVE2 complex and the regulators of small GTPase signaling, namely, *cul3a* and *racgap1*, are critical to cardiac development. A CRISPR KO of *racgap1* in zebrafish heart demonstrates atrial dilation and pericardial edema [112].

It therefore seems that *RACGAP1* could promote the proliferation of cultured cells. In samples from the right atrial wall, it can be clearly observed that the expression drops in the 30th day of culture, while for samples from the right atrial appendage, this tendency is even more significant; in the 15th day of culture, there is already 3-fold downregulation, which would suggest that, with time, possibly the cell proliferation decreases and further implies that further culture optimization is needed.

*SNAI2*, Snail Family Transcriptional Repressor 2, is an EMT-related transcription factor, promoting a migratory and invasive phenotype. Possibly the best-known target gene of this transcriptional repressor is CDH1, encoding E-cadherin, an epithelial cell adhesion molecule [113]. It is therefore not surprising that *SNAI2* plays a crucial role in developmental processes such as mesoderm formation, or neural crest migration, but also in cancer [114,115]. As for the heart development, Slug is a key for epicardial EndMT, in which a group of epicardial cells migrate to the heart and differentiate into smooth muscle cells and fibroblasts, as well as for endocardial EndMT, which is indispensable for valve development [116,117].

Apart from functions in embryonic development, *SNAI2* regulates adult stem/progenitor cell function—self-renewal, lineage commitment, and apoptosis in hematopoietic, mammary, epidermal, or mesenchymal tissues [118].

It is difficult to assess the role of *SNAI2* in the system presented here as it should consist mainly of adult porcine cardiomyocytes, not endothelial cells, or cells at the embryonic stage. In cultures derived from both heart fragments, the tendencies are relatively similar—gradual downregulation with the culture duration. If it therefore comes to any pro-angiogenic influence, it is downregulated with the culture time.

## 5. Conclusions

Primary in vitro long-term cultures of porcine cardiomyocytes isolated from two different fragments, the right atrial appendage and right atrial wall, were established and documented. The expression of potential stemness markers (GATA-4, CD44, CD90, and CD105) and CMs’ markers (α-MHC and α-actinin) in cultures was characterized using flow cytometry [54]. GATA-4, α-MHC, and α-actinin were expressed in all cultures, at all time points, while CD90 was majorly expressed at later time points [54]. It can therefore be concluded that the isolation and culture of mature cardiomyocytes was successful. CD90 and GATA-4 expression may indicate not only putative stemness, because these proteins can serve as markers of some cardiac cell types or may play roles in the functioning of mature cardiomyocytes. The influence of cell culture conditions on potential stemness properties was also characterized by transcriptomic profiling. In total, 49 genes associated with such properties demonstrated differential expression, with the 4 most deregulated genes being *SFRP2*, *PRRX1*, *RCAGP1*, and *SNAI2*.

It would be interesting to perform all the experiments after fluorescence-activated or magnetic cell sorting with properly chosen markers characteristic for the cells which have been claimed to possess self-renewal and differentiation capacities and compare them to mature cardiomyocytes and previously characterized putative cardiac progenitor cells [119,120]. Conclusions based on the presence or absence of certain stem cell markers alone cannot be treated as a proof of potency [121]. It would be crucial to establish functional assays which would enable the assessment of such features [11,12,13,120].

Lastly, the low number of cells analyzed by flow cytometry as well as the results of transcriptomic profiling, which seem to point out the presence of sources of variation in the data other than the conditions itself, clearly show that more optimization of the culture conditions is needed.

## Figures and Tables

**Figure 1 genes-14-01223-f001:**
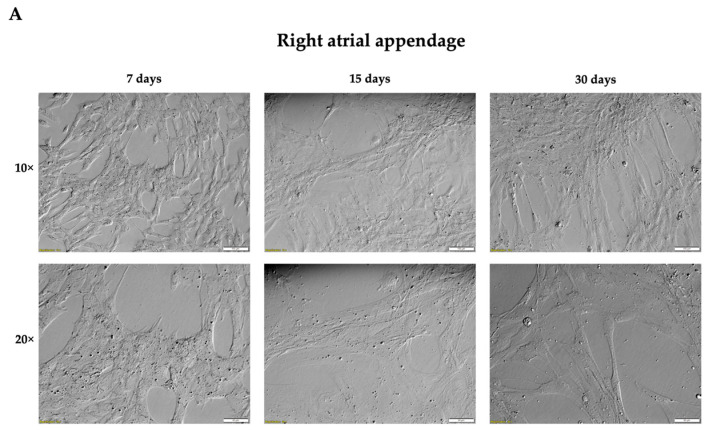
Representative images of long-term in vitro cell cultures established from 2 different heart muscle fragments—right atrial appendage (**A**) and right atrial wall (**B**). Nomarski phase/contrast images. Scale bars’ length for magnification: 10×: 100 µm, 20×: 50 µm.

**Figure 2 genes-14-01223-f002:**
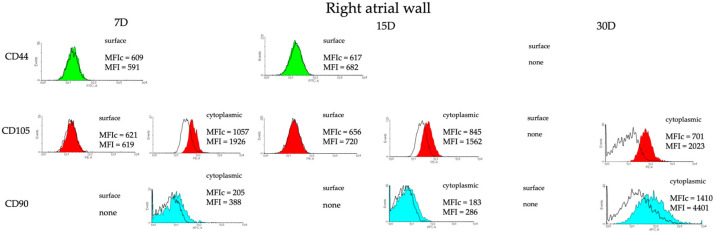
Phenotype of porcine CMs isolated from the right atrial wall in the 7th, 15th, and 30th day of culture analyzed by flow cytometry. CD44, CD105, and CD90 state for mesenchymal cell markers. MFIc—mean fluorescence intensity for control samples; MFI—mean fluorescence intensity for research samples.

**Figure 3 genes-14-01223-f003:**
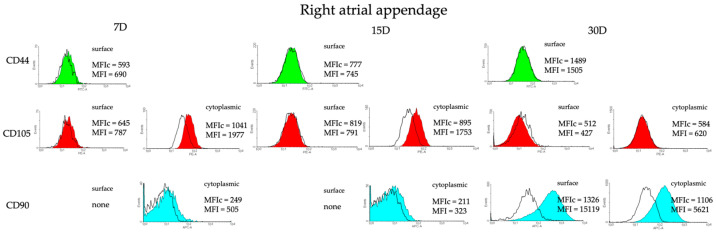
Phenotype of porcine CMs isolated from right atrial appendage in the 7th, 15th, and 30th day of culture analyzed by flow cytometry. CD44, CD105, and CD90 state for mesenchymal cell markers. MFIc—mean fluorescence intensity for control samples; MFI—mean fluorescence intensity for research samples.

**Figure 4 genes-14-01223-f004:**
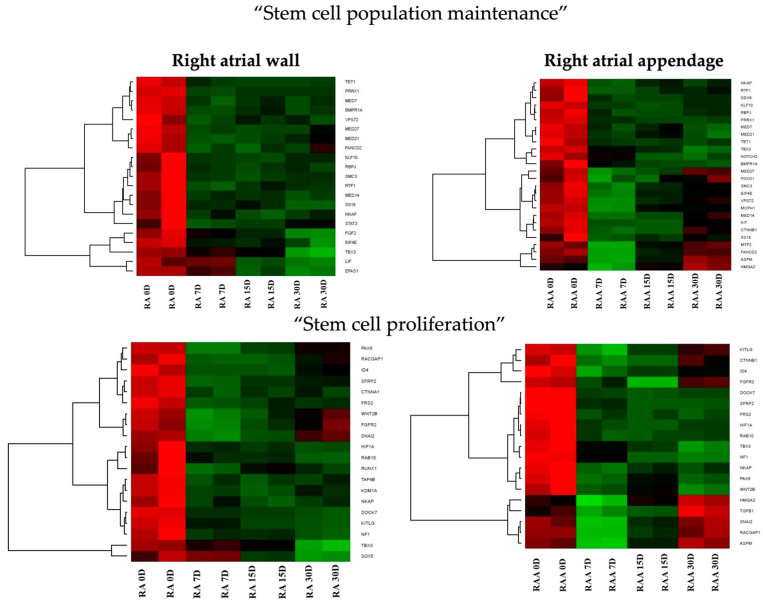
Heatmap representation of differentially expressed genes belonging to the chosen GO BP terms. The arbitrary signal intensity acquired from microarray analysis is represented by colors (green—higher expression; red—lower expression). D—day of the primary culture; RA—right atrial wall; RAA—right atrial appendage.

**Figure 5 genes-14-01223-f005:**
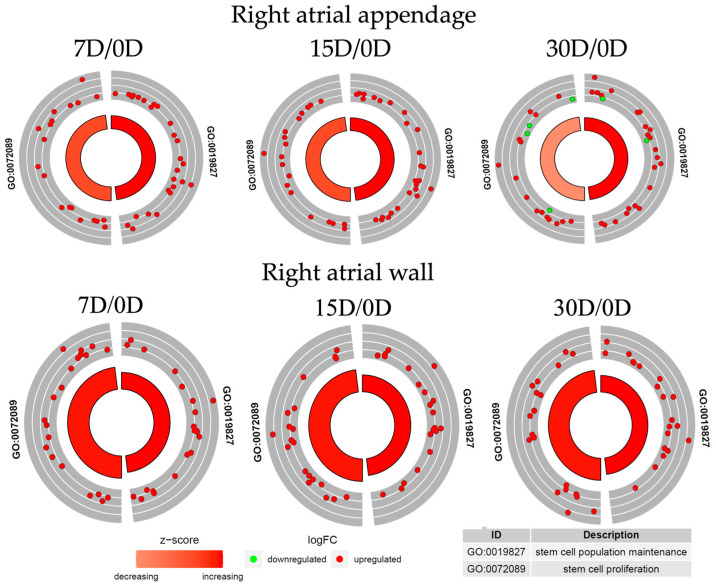
The circle plot shows the differentially expressed genes and z-score of the chosen GO BP terms “stem cell population maintenance” and “stem cell proliferation”. The outer circle shows a scatter plot for each term of the logFC of the assigned genes. Red circles display upregulation and green circles downregulation. The inner circle shows the z-score of each GO BP term. The width of each bar corresponds to the number of genes within the GO BP term and the color corresponds to the z-score. The z-score refers to the difference between the number of upregulated genes and the number of downregulated genes divided by the square root of the count, where the count is the number of genes assigned to a term.

**Figure 6 genes-14-01223-f006:**
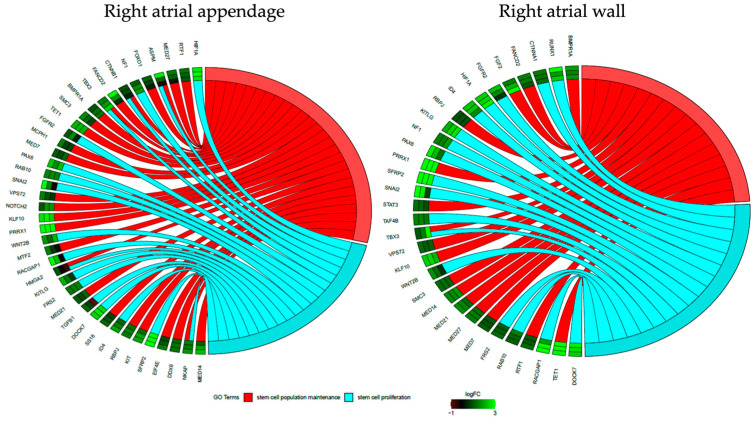
GOChord plot representing the relationship between differentially expressed genes and chosen GO BP terms: “stem cell population maintenance” and “stem cell proliferation”. The ribbons indicate which gene belongs to which categories. The color of each block closest to the gene name corresponds to the logFC of each gene (7D/0D) (green—upregulated, red—downregulated).

**Figure 7 genes-14-01223-f007:**
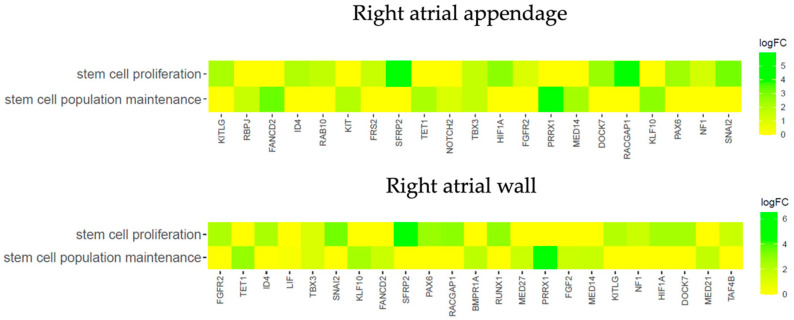
Heatmap presenting the relationship between differentially expressed genes and chosen GO BP terms: “stem cell population maintenance” and “stem cell proliferation”. The color corresponds to the logFC of a particular gene (7D/0D).

**Figure 8 genes-14-01223-f008:**
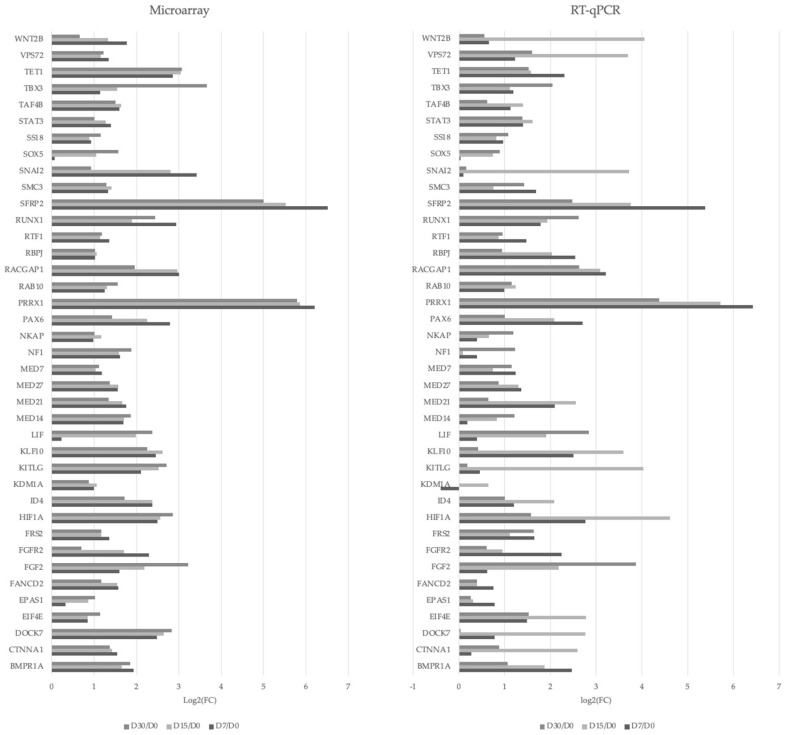
The results of the validation of the expression level of the analyzed genes by RT-qPCR presented as a bar chart. The values are presented as the log2FC of the tested culture duration vs. 0 days’ culture duration.

**Table 1 genes-14-01223-t001:** Oligodeoxynucleotides designed to detect the expression levels of the studied genes.

Gene Symbol	5′→3′ Sequence of the Forward Primer	5′→3′ Sequence of the Reverse Primer	Product Length (bp)
*ASPM*	AACAGATTACGTCGTGCTGC	CTGTCTCTAGGCCGATTCGT	205
*BMPR1A*	ATTTGGGAAATGGCTCGTCG	CCCAGCATTCCGACATTAGC	205
*CTNNA1*	CTTCTTGGCGGTCTCAGAGA	ACCCCTGGCTCATAGTTGTC	171
*CTNNB1*	CCTCTCATCAAGGCTACCGT	AGGGCTCCAGTACAACCTTC	218
*DDX6*	ACACTGCCTCAACACACTTT	GATTTCGATGTTCCTGCCTCA	220
*DOCK7*	TGCTTACTCCACCTGCATCA	ATAGTCCTTCCGCATCAGGG	198
*EIF4E*	GCAAACCTTCGGCTGATCTC	ATTAGCCATCGTCCTCCTCG	173
*EPAS1*	TGGGCTGGAGAGTTGAGAAG	AATAGTCTCCAGGGCTGCTG	151
*FANCD2*	CAGTGTATGCCGCTCCTAGA	GAGATTGCCCAGCCAGAAAG	246
*FGF2*	GCAGAAGAGAGAGGGGTTGT	CGTTCGTTTCAGTGCCACAT	195
*FGFR2*	ATCGAGATTTAGCCGCCAGA	TCCCACATTAACACCCCGAA	214
*FOXO1*	ATCACCAAGGCCATCGAGAG	AGTTCCTTCATTCTGCACGC	189
*FRS2*	TGCCTTTAAGTGTGCTCGTG	TGGGTAAGTTCTGAGCAGCA	182
*HIF1A*	CTCAGTCGTCACAGTCTGGA	CCACCTCTTTTGGCAAGCAT	212
*HMGA2*	ACCTCCCAATCTCCCGAAAG	GTTGTCCCTGGGCTGAAGT	161
*ID4*	ACTACATCCTGGACCTGCAG	CCTCCCTCTCTAGTGCTCCT	250
*KDM1A*	GTCCAGTTTGTGCCACCTCT	TGCCTACATGCCCGAACAAA	139
*KIT*	ATTGTGAATCTTCTCGGCGC	ACTCCGGGTTTCATGTCCAT	227
*KITLG*	TGGCCAGTTCTATCCATGCA	TGGTCAGGGGTAAAGGCAAA	175
*KLF10*	TGAGCTGCAGTTGGAAGTCT	TGTGAGGCTTGGCAGTATCT	246
*LIF*	GAACCTCTGAAAACTGCCGG	ACAGGAGTGATGGAAAGGGG	151
*MCPH1*	CAGCCGACCATGTTCATCAG	AGTTCTCAGAGGCACAGACC	217
*MED14*	CCACCATCCTCACTCACAGT	TCACTCCGGGTTCATTGGAA	198
*MED21*	GTCCTCCTGCCTCTTTCAGT	CCTCCAGACATGTAGCAGCT	228
*MED27*	ACTTGCATTCAGTCAACCGG	TTGTTCGACCACTTGTACGC	172
*MED7*	ACTAGTAGAAGGCACGCGAA	ACTGCCCTTCACGGTGATTA	150
*MTF2*	AAACTGCTGAGCCACCTTTG	TGCCTGGAAATGCTAGACGA	172
*NF1*	GAATCCCCACCACAGTACCA	AAGGAGATGTGGGTGTCAGG	226
*NKAP*	GAGTCCCAGGAAGAGTTGCT	CATAGCTGCACCTTCACCAG	162
*NOTCH2*	TTATGTCTCACCCCTGCCAG	ACTGTCCTGGAACGTCACAT	246
*PAX6*	TTGCCCGAGAAAGACTAGCA	GTGGGTTGTGGGATTGGTTG	196
*PRRX1*	GGACACACTACCCAGATGCT	TTTGAGGAGGGAAGCGTTCT	155
*RAB10*	ATGGCGAAGAAGACGTACGA	AGGAGGTTGTGATCGTGTGA	232
*RACGAP1*	ACCGCTGGAATACTGGAGTC	TGACAGGGAGCTGGATGAAG	184
*RBPJ*	ATGGGCAGTGGATGGAAGAA	TGTTTTGGCCGTGCAATAGT	156
*RTF1*	CCAGGCGACAGTGTAAACCT	TCGCTGGCTGACTTGGAATT	175
*RUNX1*	GTCCCAACTTCCTCTGCTCT	CTTCCACTCCGACCGACAAA	226
*SFRP2*	GCTCCAAAGGTATGTGAAGCC	GGTCTTGCTCTTGGTCTCCA	159
*SMC3*	TGGAGGACACTGAGGCAAAT	TCCTGTTGCCGCTCTAAGAA	248
*SNAI2*	GCCGAGAAGTTTCAGTGCAA	GGGTCCGAATGTGCATCTTC	169
*SOX5*	CAGCAGCAAGAACAGATCGC	AGCCAGTGTCCGTTGATCAG	147
*SS18*	CGGATATGACCAGGGACAGT	CTTGCTGCGTTTCACCTGAT	175
*STAT3*	AGCAGCAAAGAAGGAGGAGT	ACACGAGGATGTTGGTAGCA	166
*TAF4B*	GCCAGTCAGTTTCCTCAAGC	ACGAGTGTGCCAACCAATTC	227
*TBX3*	AGGGTGTTCGATGACAGACA	GACGTGGTGGTGGAGATCTT	233
*TET1*	TCTGGCAAGAAGAGAGCAGC	ATGGATGGGGTCGGTGAGTA	248
*TGFB1*	ACCATGCCAATTTCTGCCTG	GAACGCACGATCATGTTGGA	208
*VPS72*	TCCTTCGAGTACAAGAGCGG	GCACTTGCGCTTCTTATGGA	188
*WNT2B*	CAACGTGGGGACTTTGACTG	TGGCACTTACACTCCAGCTT	185
*ACTB*	CCCTGGAGAAGAGCTACGAG	CGTCGCACTTCATGATGGAG	156
*GAPDH*	CCAGAACATCATCCCTGCCT	CCTGCTTCACCACCTTCTTG	185

**Table 2 genes-14-01223-t002:** Thermal profile of RT-qPCRs.

Step		Temperature (°C)	Time (s)	Number of Cycles
Preincubation		95	600	1
Amplification	Denaturation	95	15	40
	Annealing	58	15	
	Elongation	72	15	
Melting		95	60	1
		40	60	
		70	1	
		95	1	
Cooling		37	30	1

## Data Availability

Not applicable.

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
