# Peer review of "Transcriptomic Characterization of Genes Regulating the Stemness in Porcine Atrial Cardiomyocytes during Primary In Vitro Culture"

_genes, 2023, doi:10.3390/genes14061223_

Round 1

Reviewer 1 Report

Authors in the present article tried to investigate the characterization of porcine atrial derived cardiomyocytes in vitro. the paper is well organized. However, the following points are to be considered: 

1. Please check the writing where some grammatical errors and erroneous punctuation are present, for instance, in the abstract Line 37. One point to be deleted

2. In the abstract: Provide the details for the first mentioning of the abbreviations, for example IVC, Line 38, the details of this abbreviation is missed.

3. Introduction section: line 55, the detail of HF abbreviation. If you use this abbreviation, you should mention at the first mention in each section and then unify. The same is for CPCs (Line 71) that may means Cardiac progenitor cells.

4. Methodology: unify the DMEM, the detailed description is written at the first mention only.

5. Please insert citation for any methodology performed after previously published protocols.

6. The abbreviation RT has been used for two different meanings (Room temperature, and real time PCR), please check and unify

7. In the results section, Lines 293-317; it is better to focus on the results of the present study. This paragraph fits the discussion section

8. The quality of the images is low and the font is small. Thus, information are not clear.  

9. Figure 2; the results of flowcytometry of CD 44 on day 30 are missed, please check.

10. The conclusion is very lengthy, please summarize and focus on the evidences of the present study  

The English writing is fine. However, checking for correction of some writing errors or grammatical mistakes is required.

Reviewer 2 Report

1.The text/ in Figures are too small, which is not conducive to reading.

2. The discussion and conclusion of this article is too long, especially the conclusion.

3.Why is transcriptome sequencing not considered RNA-seq? What are the advantages of using Affymetrix microarray?
